# FlowBot++: Learning Generalized Articulated Objects Manipulation via Articulation Projection

**Harry Zhang, Ben Eisner, David Held**
Robotics Institute, School of Computer Science
Carnegie Mellon University, United States
{haolunz, baeisner, dheld}@andrew.cmu.edu

**Abstract:** Understanding and manipulating articulated objects, such as doors and drawers, is crucial for robots operating in human environments. We wish to develop a system that can learn to articulate novel objects with no prior interaction, after training on other articulated objects. Previous approaches for articulated object manipulation rely on either modular methods which are brittle or end-to-end methods, which lack generalizability. This paper presents FlowBot++, a deep 3D vision-based robotic system that predicts dense per-point motion and dense articulation parameters of articulated objects to assist in downstream manipulation tasks. FlowBot++ introduces a novel per-point representation of the articulated motion and articulation parameters that are combined to produce a more accurate estimate than either method on their own. Simulated experiments on the PartNet-Mobility dataset validate the performance of our system in articulating a wide range of objects, while real-world experiments on real objects' point clouds and a Sawyer robot demonstrate the generalizability and feasibility of our system in real-world scenarios. Videos are available on our anonymized website here.

**Keywords:** Articulated Objects, 3D Learning, Manipulation

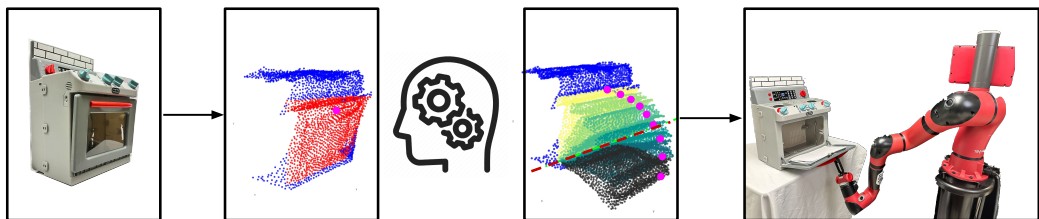

Figure 1: FlowBot++ in Action. The system first observes a point cloud observation of an articulated object and estimates the object's Articulation Flow and Articulation Projection to infer the articulation axis. Then the inferred axis is used to output a smooth trajectory to actuate the object.

## 1 Introduction

Understanding and manipulating articulated objects such as doors and drawers is a key skill for robots operating in human environments. Unlike previous work [1, 2] that learns to build the kinematic structure of an object through experience, we aim to teach a robot to manipulate novel articulated objects, transferring knowledge from prior experience with other articulated objects. While humans can manipulate novel articulated objects, constructing robotic manipulation agents that can generalize in the same way poses significant challenges, since the complex structure of such objects requires three-dimensional reasoning of their parts and functionality. Due to the large number of categories of such objects and intra-class variations of the objects' structure and kinematics, it is difficult to train perception and manipulation systems that can generalize across such variations.

7th Conference on Robot Learning (CoRL 2023), Atlanta, USA.

One approach used in previous work estimates the kinematic tree of an articulated object, using a modular pipeline of segmentation, connectivity estimation, and articulation-parameter estimation [3, 4, 5]; however, such a modular pipeline can be brittle for unseen objects, since a failure in any one component will cause failures in downstream modules. To learn a generalizable perception and manipulation pipeline, robots need to be robust to variations in the articulated objects' geometries and kinematic structures. Another recent approach [6] predicts a dense per-point estimate of how each point would move under articulated motion, without needing to predict the part's articulation parameters. Experiments demonstrate that this approach shows superior generalization to unseen articulated objects. However, the motion estimation needs to be run at each time step, which often yields jerky motion, costly computation, and poor performance in the face of heavy occlusions.

Our approach combines the advantages of both of these approaches: we jointly predict dense per-point motion of how that point would move if the object were to be articulated[1] (similar to Eisner et al. [6]) as well as a dense per-point representation of the corresponding part's articulation parameters. These two predictions are then integrated to produce a more accurate estimate of the object's motion than either method on their own. Each per-point prediction is grounded in a reference frame centered on that point, which avoids the need for the network to make a prediction relative to an unknown, global coordinate system. By estimating per-point predictions, we leverage the advantages of prior work [6] that has shown that per-point predictions enable enhanced generalization to different object geometries and kinematics.

Our system, FlowBot++, leverages these predictions to produce a smooth sequence of actions that articulate the desired part on the object. We train a ***single*** 3D perception module across many object categories; we evaluate the system on zero-shot manipulation of novel objects, without requiring additional videos or interactions with each test object. We show that the trained model generalizes to a wide variety of objects – both in seen categories as well as unseen object categories. The contributions of this paper include:

1. A novel per-point 3D representation of articulated objects that densely models the objects' instantaneous motion under actuation as well as its articulation parameters.
2. A novel approach to integrate the per-point motion prediction and per-point articulation prediction into a single prediction that outperforms either prediction individually.
3. Demonstrations of a robot manipulation system (FlowBot++) that uses the aforementioned per-point predictions to manipulate novel articulated objects in a zero-shot manner. Our experiments evaluate the performance of our system in articulating a wide range of objects in the PartNet-Mobility dataset as well as in the real-world.

## 2  Related Work

**Articulated Object Manipulation**: Manipulation of unseen articulated objects and other objects with non-rigid properties remains an open research area due to the objects' complex geometries and kinematics. Previous work proposed manipulating such objects using analytical methods, such as the immobilization of a chain of hinged objects, constraint-aware planning, and relational representations [7, 3, 8]. With the development of larger-scale datasets of articulated objects [9, 10], several works have proposed learning methods based on large-scale simulation, supervised visual learning and visual affordance learning [11, 12, 13]. Several works have focused on visual recognition and estimation of articulation parameters, learning to predict the pose [4, 5, 14, 15, 16] and identify articulation parameters [1, 17] to obtain action trajectories. Statistical motion planning methods could also be applied in the scenarios of complex hinged objects manipulation [18, 19, 20]. More closely related to our work is that of Eisner et al. [6], which learns to predict per-point motion of articulated objects under instantaneous motion. While the per-point motion affordance learning of Eisner et al. [6] outperforms previous approaches, it does not explicitly model the articulation parameters. Thus, the affordance needs to be estimated in each time step, yielding jerky robot trajectories. Concur-

---

[1]Following the convention of Eisner et al. [6], the system predicts the motion under the "opening" direction, although it can be reversed to perform closing.

rently with our work, Nie et al. [2] proposed a similar idea of estimating articulation parameters from interactions to guide downstream manipulation tasks. However, our method does not require *a priori* interactions with the manipulated objects.

**Flow for Policy Learning**: Optical flow [21] is used to estimate per-pixel correspondences between two images for object tracking and motion prediction and estimation. Current state-of-the-art methods for optical flow estimation leverage convolutional neural networks [22, 23, 24]. Dong et al. [25], Amiranashvili et al. [26] use optical flow as an input representation to capture object motion for downstream manipulation tasks. Weng et al. [27] use flow to learn a policy for fabric manipulation. Previous work generalized optical flow beyond pixel space to 3D for robotic manipulation [6, 28, 29]. We learn a 3D representation for articulated objects, *Articulation Flow*, which describes per-point correspondence between two point clouds of the same object and *Articulation Projection*, a dense projection representation of the object's articulation parameters.

## 3   Background

In this paper, we study the task of manipulating an articulated object. We assume that an articulated object could be of two classes: prismatic, which is parameterized by a translational axis, or revolute, which is parameterized by a rotational axis. Eisner et al. [6] derived the optimal instantaneous motion to articulate objects using physics and we review their reasoning here to better motivate our method. Suppose we are able to attach a gripper to some point $p \in \mathcal{P}$ on the surface $\mathcal{P} \subset \mathbb{R}^3$ of a child link with mass m. At this point, the policy can apply a 3D force $\mathbf{F}$, with constant magnitude $||\mathbf{F}||$ to the object at that point. We wish to choose a contact point and force direction that maximize the acceleration of the articulation's child link. Both articulation types are parameterized via axes $\mathbf{v}(t) = \boldsymbol{\omega} t + v$, where $\boldsymbol{\omega}$ is a unit vector that represents the axis direction and $v$ represents the origin of the axis[2] (dashed lines in Fig. 2). Based on Newton's Second Law, Eisner et al. [6] derived that for *prismatic joints*, a force $\mathbf{F}$ maximizes the articulated part's acceleration if the force's direction is parallel to the part's axis $\boldsymbol{\omega}$; the force's point of exertion does not matter as long as the direction is parallel to $\boldsymbol{\omega}$. For *revolute joints*, a force $\mathbf{F}$ maximizes the articulation part's acceleration when $\mathbf{F}$ is tangent to the circle defined by axis $\mathbf{v}$ and radius $\mathbf{r}$, which connects the point of exertion to the nearest point on the axis. Selecting the point that maximizes $\mathbf{r}$ produces maximal linear acceleration. Thus for revolute joints, the optimal choice for the force $\mathbf{F}$ is to pick a point on the articulated part that is farthest from the axis of rotation $\mathbf{v}$ and apply a force tangent to the circle defined by $\mathbf{r}$ and axis $\mathbf{v}$. Similar to Eisner et al. [6], we predict each point's **Articulation Flow**, defined as follows: for each point $p$ on each link on the object, define a vector $f_p$ in the direction of the motion of that point caused by an infinitesimal displacement $\delta\theta$ of the joint in the opening direction, normalized by the largest such displacement on the link:

$$\text{Articulation Flow: } f_p = \begin{cases} \boldsymbol{\omega}, & \text{if prismatic} \\ \frac{\boldsymbol{\omega} \times \mathbf{r}}{||\boldsymbol{\omega} \times \mathbf{r}_{\max}||} & \text{if revolute} \end{cases} \tag{1}$$

where $\mathbf{r}$ is the radius for a contact point and $\mathbf{r}_{\max}$ represents the longest radius to the axis on a revolute object. Eisner et al. [6] showed how the predicted Articulation Flow can be used to derive a policy to manipulate previously unseen articulated objects. Below, we will treat the notation for the articulation axis $\mathbf{v}(t)$ and $\mathbf{v}$ interchangably.

## 4   FlowBot++: From New Representations to Smooth Trajectories

Eisner et al. [6] predicts the per-point Articulation Flow, which allows the method to avoid needing to directly predict the overall kinematic structure. However, the Articulation Flow needs to be re-predicted each timestep, since it captures the ideal instantaneous motion of the articulated part under an infitesimal displacement $\delta\theta$; after the robot takes a small action, the Articulation Flow

---

[2]There could be infinitely many possible prismatic axes, but the convention of CAD models is defining an axis through the centroid of the prismatic part.

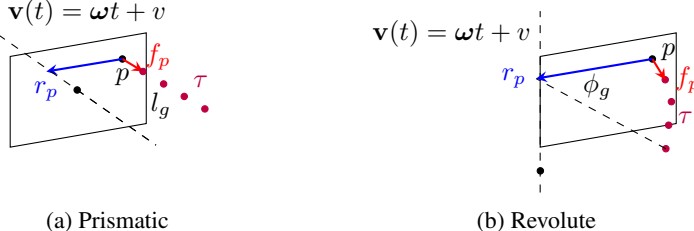

| (a) Prismatic | (b) Revolute |

Figure 2: For each point $p$ on the object, Articulation Flow $f_p$ [6] represents its instantaneous motion under a force in the opening direction; our new representation, Articulation Projection $r_p$, represents the displacement that projects $p$ to the articulation axis $\mathbf{v}$. We train a network to predict both $f_p$ and $r_p$ and combine their predictions to get a smoother and more robust estimate. The purple points represent an interpolated prismatic trajectory of length $l_g$ and an interpolated revolute trajectory of $\phi_g$ angle rotation. This corresponds to the trajectory prediction described in Sec. 4.2.

$f_p$ for revolute joints will change direction, since the direction of motion caused by an infitesimal displacement $\delta\theta$ will have changed. Since the motion derived from Articulation Flow by construction only models small movements, this representation lacks the ability to predict a full, multi-step trajectory from the initial observation.

## 4.1 A New Representation of Articulated Objects

In order to overcome the limitations of previous work [6, 12], if we could estimate the articulation parameters $\mathbf{v}$ (or equivalently, $\boldsymbol{\omega}$ and $v$), then we would be able to derive the full trajectory of any arbitrary point $p$ on the object would move under articulation of the object under kinematic constraints of $\mathbf{v}$. To achieve this, we formally define a 3D dense visual representation of articulation parameters on top of Articulation Flow; we illustrate each representation graphically in Fig. 2. In this work, we introduce another 3D representation that densely represents the object's articulation parameters, **Articulation Projection** - for each point on the articulated part of the object, we define a vector $r_p$ that represents the displacement from the point itself to its projection onto the part's articulation axis $\mathbf{v}(t)$. For each point $p \in \mathbb{R}^3$ of articulated object $\mathcal{P}$, we define the Articulation Projection $r_p$ mathematically as follows:

$$\text{Articulation Projection: } r_p = \text{proj}_{\mathbf{v}} p - p = \left(\boldsymbol{\omega}\boldsymbol{\omega}^T - \mathbf{I}\right)(p - v) \tag{2}$$

where $\boldsymbol{\omega}$ is a unit vector that represents the axis direction and $v$ represents the origin of the axis and $\mathbf{I}$ represents a 3x3 identity matrix. Mathematically, $r_p$ is calculated as the difference between the vector $\overrightarrow{vp}$ and its projection onto $\mathbf{v}$, which is equivalently *a perpendicular vector from each point to the axis of rotation defined by direction $\boldsymbol{\omega}$ and origin $v$*, which is represented using the blue vectors in Fig. 2. As we will see, Articulation Projection can be used to predict a longer articulation trajectory while still inheriting the generalization properties of prior work [6].

## 4.2 Manipulation via Learned Articulation Flow and Articulation Projection

As mentioned above, previous work [6] used the Articulation Flow $f_p$ to compute the instantaneous motion direction, which needs to be re-predicted every time step. In contrast, if we were able to infer the articulation parameters (rotation axis $\boldsymbol{\omega}$ and its origin $v$), we could then analytically compute a multi-step articulation trajectory using the initial observation of the object without the need to stop and re-compute each step to actuate the articulated part as in previous work [6]. However, Eisner et al. [6] has shown that estimating the Articulation Flow results in more generalizable prediction than estimating the articulation parameters directly. We propose to estimate an articulation trajectory from the Articulation Projection. Because the Articulation Projection is a dense per-point prediction with a reference frame defined at each point (similar to Articulation Flow [6]), it inherits the generalization properties of Articulation Flow that allows it to be used for unseen articulated objects, thus hopefully obtaining the best of both worlds. Further, we will combine both predictions in Section 4.3 to obtain an even more accurate prediction.

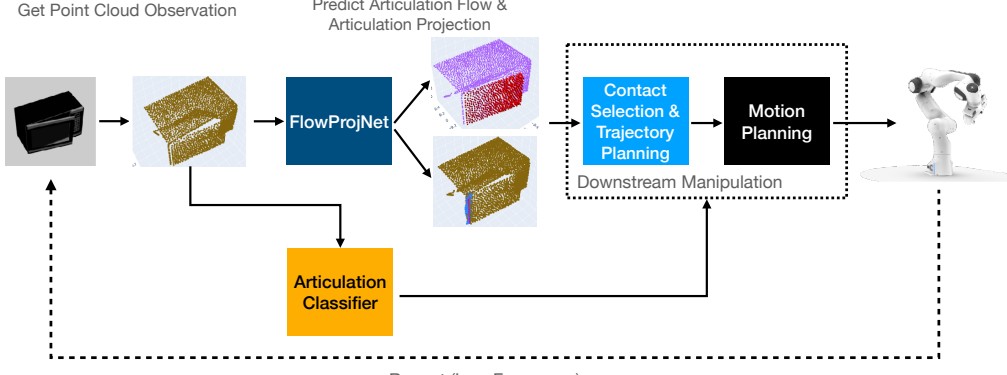

Figure 3: FlowBot++ System Overview. Our system in deployment first takes as input a partial point cloud observation of an articulated object (a microwave shown here) and uses FlowProjNet to jointly estimate the object's Articulation Flow (**top**) and Articulation Projection (**bottom**, points displaced by AP shown). The estimates will then be used in the downstream manipulation pipeline that interpolates and follows the planned trajectory smoothly. Unlike FlowBot3D [6], we do not repeat the estimation every step. Instead, we repeat this loop in a much lower frequency (once every H steps) to improve the smoothness of the planned trajectory.

For *revolute objects*, we infer the rotation axis using both the predicted Articulation Flow (Eq. 1) as well as the Articulation Projection (Eq. 2). We then estimate an $H$-step trajectory of the contact point under articulated motion in the opening direction. To obtain the articulation parameter $\boldsymbol{\omega}$, we calculate the cross product between the articulation flow $f_p$ and the articulation projection $r_p$ respectively as the articulation axis $\boldsymbol{\omega}$ should be perpendicular to both vectors. To compute the origin of the rotation, we use the point displaced by the articulation projection vector. For each point $p$, $p + r_p$ projects the point onto the rotation axis $\mathbf{v}$, effectively giving us an estimate of the origin of the rotation:

$$\hat{\boldsymbol{\omega}} = \frac{r_p \times f_p}{||r_p \times f_p||}, \quad \hat{v} = p + r_p \tag{3}$$

These predictions of $\hat{v}$ are visualized as the blue points in the bottom output branch of Fig. 3. Using the inferred axis direction $\hat{\boldsymbol{\omega}}$ and origin $\hat{v}$, we can interpolate a smooth trajectory: given a contact point $p$, the proposed trajectory should lie on a circle in a plane perpendicular to the inferred axis parameterized by the normalized direction $\hat{\boldsymbol{\omega}}$ and the origin $\hat{v}$. Formally, with Rodrigues' Formula of Lie algebra, given an angle of rotation $\phi$ about the normalized vector $\hat{\boldsymbol{\omega}}$ we are able to define a rotation matrix as follows:

$$\mathbf{R}(\phi) = \mathbf{I} + \sin\phi[\hat{\boldsymbol{\omega}}]_\times + (1 - \cos\phi)[\hat{\boldsymbol{\omega}}]_\times^2 \tag{4}$$

where $\mathbf{I}$ is identity matrix and $[\hat{\boldsymbol{\omega}}]_\times$ is the skew-symmetric matrix of $\hat{\boldsymbol{\omega}}$, and the rotated point $p$ about this inferred axis by angle $\phi$ is given by: $p' = \mathbf{R}(\phi)(p - \hat{v}) + \hat{v}$. In order to obtain a trajectory, we interpolate $K$ angles between 0 and a goal angle $\phi_g$, and the smooth, $K$-step trajectory[3] that rotates the contact point $p$ by angle $\phi_g$ about the estimated axis parameterized by $\hat{\boldsymbol{\omega}}$ and $\hat{v}$ is calculated via:

$$\tau_{\text{revolute}} = \left\{ \mathbf{R}\left(\frac{i}{K}\phi_g\right)(p - \hat{v}) + \hat{v} \right\}_{\forall i \in [0,K]} \tag{5}$$

We learn a separate articulation type classifier $f_\psi$ which takes as input the point cloud observation of the object and classifies if the object is prismatic or revolute to guide the trajectory planning step. For *prismatic objects*, since the motion direction and the axis direction should be the same, we just use the normalized flow prediction $f_p$ to compute the prismatic articulation axis, and we use the estimated axis to compute a smooth, $K$-step trajectory indexed by $i$ to translate the contact point $p$

---

[3]In practice, the goal translation distance and goal angle is set large enough so that the termination condition (part being fully open) of the policy can be triggered.

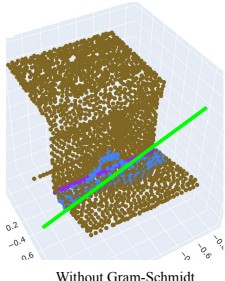
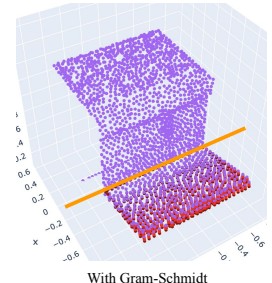
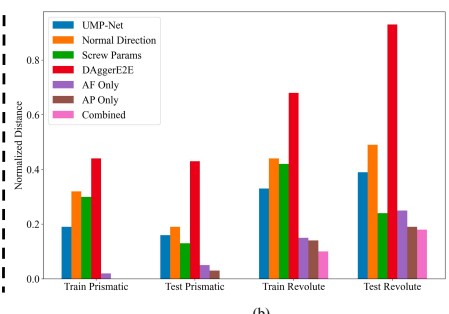

Without Gram-Schmidt  With Gram-Schmidt

(a)  (b)

Figure 4: Gram-Schmidt Correction & Performance of Different Methods. In **(a)**, without Gram-Schmidt, the inferred articulation axis $\hat{\boldsymbol{\omega}}$ (green) is not accurate; blue points show the displacement of points by the Articulation Projection $r_p$. Using Gram-Schmidt, we use the Articulation Flow $f_p$ (red vectors) to correct the axis direction; $\hat{\boldsymbol{\omega}}$ is now perpendicular to the Articulation Flow and aligns better with the ground-truth axis direction. In **(b)**, we show the bar plot of our method compared with several baseline methods on training and testing prismatic/revolute objects using normalized distance ($\downarrow$). Note the performance gain after correction via Gram-Schmidt (AP Only vs Combined). Some values are not visible because they are $< 0.05$.

by a goal translation distance $l_g$:

$$\hat{\boldsymbol{\omega}} = \frac{f_p}{||f_p||}, \quad \tau_{\text{prismatic}} = \left\{ p + \frac{i}{K} l_g \hat{\boldsymbol{\omega}} \right\}_{i \in [0,K]} \tag{6}$$

In practice, to make the prediction more robust, we use a segmentation mask to average the predictions over all points in a given part to get an aggregated version of $\hat{\boldsymbol{\omega}}$ and $\hat{v}$. We also deploy an MPC-style controller that replans after taking the first $H$ steps of trajectory for better performance.

### 4.3 Jointly Learning Articulation Flow and Articulation Projection

Empirically, the Articulation Projection prediction is sometimes less accurate than the Articulation Flow prediction. On the other hand, the Articulation Flow represents the instantaneous motion and thus needs to predicted at each timestep, which prevents multi-step planning. To obtain the best of both worlds, we correct the Articulation Projection $r_p$ prediction with the flow estimate using Gram-Schmidt on $r_p$ to make it perpendicular to the Articulation Flow $f_p$; the correction is computed as: $\tilde{r}_p = r_p - \text{proj}_{f_p} r_p$, which is used in place of $r_p$ in Eq. 3. The effect of using Gram-Schmidt is shown in Fig. 4a.

We learn to estimate the Articulation Flow $f_p$ and the Articulation Projection $r_p$ jointly using a deep neural network, which we refer to as FlowProjNet, denoted as $f_\theta$. We assume that the robot has a depth camera and records point cloud observations $O_t \in \mathbb{R}^{3 \times N}$, where $N$ is the total number of observable points from the sensor. The task is for the robot to articulate a specified part through its entire range of motion. For each configuration of the object, there exists a unique ground-truth Articulation Flow $\mathbf{f}_t$ and ground-truth Articulation Projection $\mathbf{r}_t$, given by Equations 1 and 2. Thus, our learning objective is to find a function $f_\theta(O_t)$ that predicts Articulation Flow and Articulation Projection directly from point cloud observations. We define the objective of minimizing the joint L2 error of the predictions:

$$\mathcal{L}_{\text{MSE}} = ||(\mathbf{f}_t \oplus \mathbf{r}_t) - f_\theta(O_t)||_2^2 \tag{7}$$

where $\oplus$ represents concatenation. We train FlowProjNet via supervised learning with this loss. We describe the method to obtain ground-truth labels in Appendix E. We implement FlowProjNet using a segmentation-style PointNet++ [30] as a backbone, and we train a *single* model across all categories, using a dataset of synthetically-generated data tuples based on the ground-truth kinematics provided by the PartNet-Mobility dataset [10]. Details of our dataset construction and model architecture can be found in Appendix E.

## 5 Experiments

We conduct a wide range of simulated and real-world experiments to evaluate FlowBot++.

| | Novel Instances in Train Categories | | | | | | | | | | | | Test Categories | | | | | | | | | | |
|---|---|---|---|---|---|---|---|---|---|---|---|---|---|---|---|---|---|---|---|---|---|---|---|
| | AVG. | | | | | | | | | | | | AVG. | | | | | | | | | | |
| UMP-Net [12] | 0.18 | 0.18 | 0.17 | 0.32 | 0.32 | 0.05 | 0.06 | **0.12** | 0.24 | 0.23 | 0.18 | 0.08 | 0.15 | 0.23 | 0.14 | 0.04 | 0.00 | 0.25 | 0.27 | 0.09 | 0.21 | **0.13** | 0.19 |
| Normal Direction | 0.41 | 0.52 | 0.67 | 0.16 | 0.19 | 0.51 | 0.60 | 0.13 | 0.14 | 0.45 | 0.71 | 0.31 | 0.40 | 0.79 | 0.43 | 0.07 | 0.00 | 0.64 | 0.15 | 0.70 | 1.00 | 0.41 | 0.29 |
| Screw Parameters [1] | 0.42 | 0.44 | 0.42 | 0.44 | 0.16 | 0.64 | 0.52 | 0.21 | 0.57 | 0.43 | 0.58 | 0.11 | 0.24 | 0.28 | 0.25 | 0.14 | 0.12 | 0.26 | 0.12 | 0.11 | 0.12 | 0.32 | 0.26 |
| DAgger E2E [31] | 0.52 | 0.25 | 0.86 | 0.51 | 0.71 | 0.55 | 1.00 | 0.86 | 0.80 | 0.74 | 0.58 | 0.40 | 0.64 | 0.53 | 0.52 | 1.00 | 0.65 | 0.75 | 0.54 | 0.63 | 0.68 | 0.87 | 0.74 |
| FlowBot3D (AF Only) [6] | 0.17 | 0.42 | 0.22 | 0.16 | 0.17 | 0.03 | **0.00** | 0.20 | 0.51 | 0.07 | **0.00** | 0.08 | 0.21 | 0.17 | 0.29 | **0.00** | 0.06 | **0.21** | 0.10 | **0.06** | 0.16 | 0.29 | 0.73 |
| AP Only (Ours) | 0.11 | 0.07 | 0.01 | 0.05 | 0.08 | 0.06 | 0.22 | 0.17 | 0.15 | 0.14 | 0.13 | 0.08 | 0.13 | 0.12 | 0.18 | 0.09 | 0.00 | 0.29 | 0.02 | 0.14 | 0.07 | 0.21 | 0.27 |
| **FlowBot++ (Ours - Combined)** | **0.07** | **0.04** | **0.01** | **0.04** | **0.08** | **0.02** | 0.19 | 0.17 | **0.14** | **0.07** | 0.09 | **0.02** | **0.10** | **0.00** | **0.11** | 0.09 | **0.00** | 0.23 | **0.02** | 0.09 | **0.09** | 0.20 | **0.18** |

Table 1: Normalized Distance Metric Results (↓): Normalized distances to the target articulation joint angle after a full rollout across different methods. The lower the better.

## 5.1 Simulation Results

To evaluate our method in simulation, we implement a suction gripper in PyBullet. We consider the same subset of PartNet-Mobility as in previous work [12, 6]. Each object starts in the "closed" state (one end of its range of motion), and the goal is to actuate the joint to its "open" state (the other end of its range of motion). During our experiments, we use the same Normalized Distance metric defined in prior work [6, 12].

**Baseline Comparisons**: We compare our proposed method with several baseline methods: UMP-Net [12], Normal Direction, Screw Parameters [1], DAgger End2End [31], FlowBot3D (AF Only) [6] which only uses **A**rticulation **F**low, and our model without the Gram-Schmidt correction ("AP Only"), which only uses the inferred **A**rticulation **P**arameters. Please refer to Appendix B for more details. Each method above consists of a single model trained across all PartNet-Mobility training categories.

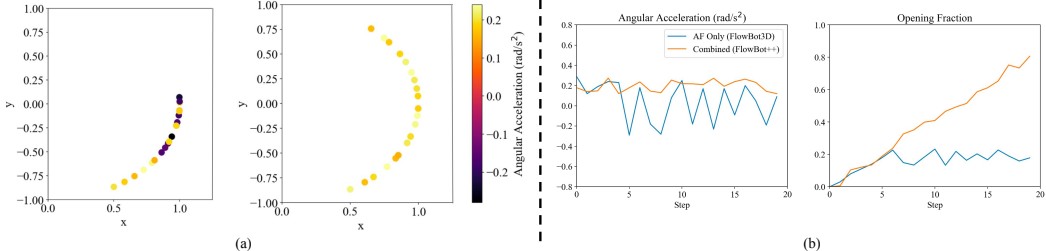

(a)                                                                      (b)

Figure 5: Comparison of Angular Acceleration and Opening Trajectory between FlowBot3D and FlowBot++. In (**a**), we show two 20-step trajectories' scatter plots of the contact point on a revolute object using FlowBot3D (left) and FlowBot++ (right), colored using the signed intensity of the angular acceleration. In (**b**), we plot 20 steps vs. the average angular acceleration and opening fraction across 300 trials involving 15 revolute objects. Both plots show that FlowBot++ is able to produce more consistent motions and open the objects further under the same number of steps.

**Analysis**: Results are shown in Table 1; please refer to Appendix D for comparisons using other metrics. Overall, FlowBot++ outperforms all of the baselines, including FlowBot3D [6] in terms of the normalized distance across many object categories, including unseen test categories. Moreover, without Gram-Schmidt correction (AP Only), the performance degrades but is still slightly better than FlowBot3D [6]. To illustrate why, we show an example in Fig. 5a in which we plot the $xy$ locations of the contact point in the first 20 execution steps as well as the angular acceleration when using each method for articulation. As shown, both FlowBot3D and FlowBot++ perform reasonably well in the first 5 steps due to the absence of occlusions. However, after 5 steps, when the robot starts to heavily occlude the object, FlowBot3D's prediction of each step begins to make the contact point go back and forth, yielding little progress in the opening direction. In contrast, FlowBot++ plans a longer, multi-step trajectory at the start of the motion, interpolated via Eq. 5. This trajectory is much more consistent and smooth in terms of the direction of the motion. This trend is further shown in Fig. 5b, where we show the average angular acceleration and opening fraction across 15 revolute objects that were challenging for FlowBot3D [6]; FlowBot++ is able to achieve a larger

opening fraction and a smoother trajectory due to its lower replanning frequency, which also leads it to suffer less from occlusions since it can make a longer-scale prediction before contact.

**Execution Time Comparisons**: Here, we provide a comparison of execution time to open objects between FlowBot3D and FlowBot++. In simulation, under the same setup, execution wall-clock time gets reduced from 17.1 seconds per object on average to 1.2 seconds per object on average. The reason for the increased speed is twofold: First, FlowBot++ deploys an MPC-style controller, which only replans after $H$ steps ($H = 7$ in our experiments), compared to FlowBot3D's closed-loop controller which requires replanning every step, since only the instantaneous motion is predicted. Second, with fewer replanning steps, FlowBot++ is less prone to be affected by occlusions because, in each replanning step, it rolls out a longer trajectory, as opposed to FlowBot3D, in which occlusions inevitably affect each step's prediction quality. The reduced prediction quality of FlowBot3D causes the robot to move the object back and forth, extending the execution time.

### 5.2 Real-World Experiments

We conduct qualitative real-world experiments to assess the sim2real capabilities of Flow-Bot++. We run FlowBot++ trained in simulation directly on denoised point clouds collected on real-world objects [32] without any retraining. We test our method on 6 real-world prismatic (Drawer) and revolute (Fridge, Oven, Toilet, Microwave, Trashcan) objects. In real-world trials, we provide hand-labeled segmentation masks to the network, which are used to filter and aggregate the results. We use a heuristically-defined grasping policy to select a grasp point and execute the planned trajectories

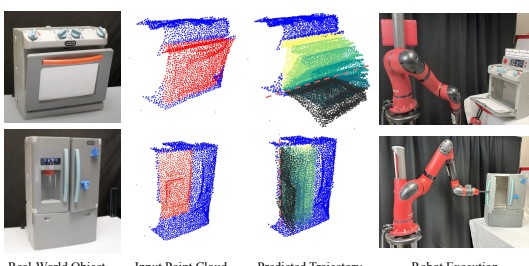

Real-World Object  Input Point Cloud  Predicted Trajectory  Robot Execution

Figure 6: Real-World Experiments. We show two examples of FlowBot++ trained in simulation predicting and executing a full opening trajectory on real-world objects without any fine-tuning or retraining.

on a Sawyer Black Robot equipped with a parallel-jaw gripper. Following the robot controller design in [28], once a contact point is chosen, we could transform the contact point using Eq. 5 because the gripper is now modeled as rigidly attached to the contact point, forming a full robot end-effector trajectory. Example predictions are shown in Fig. 6, in which all points on the segmented part are collectively transformed using Eq. 5, showing a full trajectory of the articulation. Real-world experiments show promising results that our networks transfer the learned 3D geometry information to real-world data. We show side-by-side comparisons between FlowBot++ and FlowBot3D on our website. FlowBot3D [6] completely failed on real-world oven, while FlowBot++ is able to predict a feasible full-trajectory to open the oven door with only one planning step. We also observe that FlowBot++ is able to produce smoother trajectories without undesired motions and movements of the objects and finish the execution in a shorter time frame, corroborating our findings in simulation. Details of the real-world experiments and system implementation are documented in Appendix F.

## 6 Conclusions and Limitations

In this work, we propose a novel visual representation for articulated objects, namely Articulation Projection, as well as a policy, FlowBot++, which leverages this representation, combined with Articulation Flow, to successfully manipulate articulated objects, outperforming previous state-of-the-art methods. We demonstrate the effectiveness of our method in both simulated and real environments and observe strong generalization to unseen objects.

**Limitations:** While our method shows strong performance on a range of object classes, there is still room for improvement. When both Articulation Flow and Articulation Projection predictions are incorrect, the predicted trajectory cannot be further corrected, causing failures. Furthermore, in order to aggregate all the per-point predictions on a part, a segmentation mask is required, adding another potential point of failure. Still, this work represents a step forward in the manipulation of unseen articulated objects, and we hope it provides a foundation for future work in this direction.

**Acknowledgments**

This material is based upon work supported by the National Science Foundation under Grant No. IIS-1849154. We are grateful to Prof. Daniel Seita for his helpful feedback and discussion on the paper.

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

# Appendix

Results are better shown visually in videos. Please refer to our website for video results.

## Table of Contents

## A   Full FlowBot++ Manipulation Policy

Given an articulated object, we first observe an initial observation $O_0$, which is used to classify the object's articulation type. We then predict the initial flow $f_0$ and projection $r_0$, where $f_0$ is used to select a contact pose and grasp object. Then the system infers the articulation parameters based on Eq. 5 or 6 and follows the first $H$ steps. This process repeats in a low-frequency if re-planning is needed until the object has been fully-articulated, a max number of steps has been exceeded, or the episode is otherwise terminated. See Algorithm 1 for a full description of the generalized policy. The while loop runs in a much lower frequency compared to FlowBot3D, which further bypasses the potential error from heavy occlusions.

## B   Ablations

We document a variety of Ablation Studies in this section. Specifically, we investigate the effect of using different $H$ values (i.e. replanning frequency), Gram-Schmidt Correction, and mean aggregation using part segmentation masks.

### B.1   Controller $H$ Values (Replanning Frequency)

$H$ values represent how many steps of the interpolated trajectory we aim to execute after each prediction. Thus, it also represents the replanning frequency, where s higher $H$ value means a lower replanning frequency, and vice versa.     As shown in Fig. 7 and Table 2, when the replanning

**Algorithm 1** The FlowBot++ articulation manipulation policy

---

**Require:** $\theta \leftarrow$ parameters of a trained flow-projection prediction network, $H \leftarrow$ controller lookahead horizon, $\psi \leftarrow$ articulation type classifier parameters
    $O_0 \leftarrow$ Initial observation
    $\texttt{artType} \leftarrow f_\psi(O_0)$, Classify articulation type
    $f_0, r_0 \leftarrow f_\theta(O_0)$, Predict the initial flow and projection
    $g_0 = \texttt{SelectContact}(O_0, f_0)$, Select a contact pose and grasp object as shown in [6]
    $done \leftarrow$ False
    **while** not $done$ **do**
        $O_t \leftarrow$ Observation
        $f_t, r_t \leftarrow f_\theta(O_t)$, Predict the current articulation flow and articulation projection
        $\tau_t \leftarrow \texttt{TrajCalculation}(f_t, r_t)$, Calculate trajectory using Eq. 5 or 6 based on $\texttt{artType}$
        Follow the first $H$ steps in $\tau_t$ (MPC)
        $done \leftarrow \texttt{EpisodeComplete}()$

---

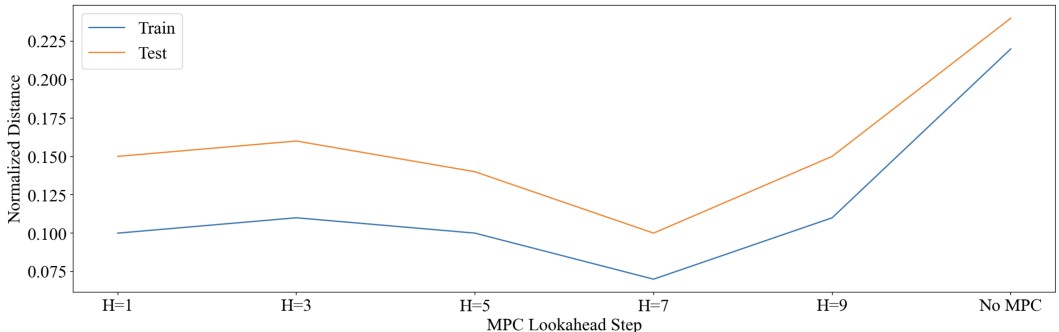

Figure 7: Ablation Studies on Lookahead Horizon. The plot shows the normalized distance performances of different $H$ values on both train and testing objects.

frequency is too low or too high, the performance becomes suboptimal. When $H = 1$, the system effectively reduces to FlowBot 3D, which replans every step. Another interesting comparison in this experiment is that when we do not use this MPC controller (i.e. we do not replan and trust the one-shot open-loop plan), the performance degrades by a lot. This suggest that we do need to replan at a certain frequency to correct ourselves. Experiments suggest that the optimal $H$ value here is $H = 7$ and we use this value in our final system.

## B.2 Mean Aggregation with Segmentation Masks

We also ablate on the choice of using a segmentation mask to aggregate the articulation parameters estimates in Table 3. Results suggest the effectiveness of using segmentation masks to aggregate multiple results for a robust estimate. Such effectiveness is better shown on revolute objects as flow directions alone suffice to produce a good motion for prismatic objects.

## C Baselines

**Baseline Comparisons**: We compare our proposed method with several baseline methods:

- **UMP-Net**: This is the same implementation/code base as provided in [12].

- **Normal Direction**: We use off-the-shelf normal estimation to estimate the surface normals of the point cloud using Open3D [33]. To break symmetry, we align the normal direction vectors to the camera. At execution time, we first choose the ground-truth maximum-flow point and then follow the direction of the estimated normal vector of the surface.

Table 2:

| | Novel Instances in Train Categories | | | | | | | | | | | | Test Categories | | | | | | | | | | |
|---|---|---|---|---|---|---|---|---|---|---|---|---|---|---|---|---|---|---|---|---|---|---|---|
| | AVG. | | | | | | | | | | | | AVG. | | | | | | | | | | |
| FlowBot++ (H=1) | 0.10 | 0.06 | 0.09 | 0.09 | 0.09 | 0.02 | 0.01 | 0.17 | 0.20 | 0.17 | 0.15 | 0.02 | 0.15 | 0.00 | 0.10 | 0.12 | 0.00 | 0.33 | 0.12 | 0.22 | 0.17 | 0.18 | 0.25 |
| FlowBot++ (H=3) | 0.11 | 0.03 | 0.10 | 0.06 | 0.09 | 0.00 | 0.24 | 0.19 | 0.18 | 0.21 | 0.09 | 0.03 | 0.16 | 0.15 | 0.14 | 0.14 | 0.00 | 0.34 | 0.04 | 0.21 | 0.20 | 0.19 | 0.27 |
| FlowBot++ (H=5) | 0.10 | 0.06 | 0.08 | 0.06 | 0.09 | 0.01 | 0.23 | 0.17 | 0.17 | 0.12 | 0.11 | 0.02 | 0.14 | 0.12 | 0.09 | 0.15 | 0.00 | 0.27 | 0.01 | 0.16 | 0.18 | 0.18 | 0.22 |
| FlowBot++ (H=7) | 0.07 | 0.04 | 0.01 | 0.04 | 0.08 | 0.02 | 0.19 | 0.17 | 0.14 | 0.07 | 0.09 | 0.02 | 0.10 | 0.00 | 0.11 | 0.09 | 0.00 | 0.23 | 0.02 | 0.09 | 0.09 | 0.20 | 0.18 |
| FlowBot++ (H=9) | 0.11 | 0.06 | 0.05 | 0.05 | 0.27 | 0.03 | 0.17 | 0.19 | 0.25 | 0.06 | 0.07 | 0.02 | 0.15 | 0.13 | 0.09 | 0.22 | 0.00 | 0.14 | 0.22 | 0.11 | 0.21 | 0.17 | 0.16 |
| FlowBot++ (no MPC) | 0.22 | 0.25 | 0.24 | 0.20 | 0.50 | 0.04 | 0.22 | 0.32 | 0.37 | 0.10 | 0.09 | 0.07 | 0.24 | 0.31 | 0.14 | 0.35 | 0.00 | 0.21 | 0.32 | 0.22 | 0.18 | 0.22 | 0.40 |

Table 2: Ablation Studies of Lookahead Horizon ($H$) via Normalized Distance ($\downarrow$). The lower the better.

| | Novel Instances in Train Categories | | | | | | | | | | | | Test Categories | | | | | | | | | | |
|---|---|---|---|---|---|---|---|---|---|---|---|---|---|---|---|---|---|---|---|---|---|---|---|
| | AVG. | | | | | | | | | | | | AVG. | | | | | | | | | | |
| FlowBot++ | 0.07 | 0.04 | 0.01 | 0.04 | 0.08 | 0.02 | 0.19 | 0.17 | 0.14 | 0.07 | 0.09 | 0.02 | 0.10 | 0.00 | 0.11 | 0.09 | 0.00 | 0.23 | 0.02 | 0.09 | 0.09 | 0.20 | 0.18 |
| FlowBot++ No Seg | 0.10 | 0.06 | 0.01 | 0.12 | 0.08 | 0.09 | 0.21 | 0.18 | 0.14 | 0.09 | 0.09 | 0.08 | 0.13 | 0.15 | 0.15 | 0.09 | 0.00 | 0.25 | 0.02 | 0.12 | 0.09 | 0.19 | 0.25 |

Table 3: Ablation Studies of Mean Aggregation with Segmentation via Normalized Distance ($\downarrow$). The lower the better.

- **Screw Parameters**: We predict the screw parameters for the selected joint of the articulated object. We then generate 3DAF from these predicted parameters and use the FlowBot3D policy on top of the generated flow.

- **DAgger E2E**: We also conduct behavioral cloning experiments with DAgger [31] on the same expert dataset as in the BC baseline. We train it end-to-end (E2E), similar to the BC model above.

- **FlowBot3D**: We also call this AF Only, since it only uses the articulation flow $f_p$ during planning) [6],

- **Without Gram-Schmidt Correction**: Also called AP Only. This is our model but without Gram-Schmidt correction via $f_p$, hence the name AP Only, since it only uses the inferred articulation parameters during planning without $f_p$ correction).

| | Novel Instances in Train Categories | | | | | | | | | | | | Test Categories | | | | | | | | | | |
|---|---|---|---|---|---|---|---|---|---|---|---|---|---|---|---|---|---|---|---|---|---|---|---|
| | AVG. | | | | | | | | | | | | AVG. | | | | | | | | | | |
| UMP-Net [12] | 0.73 | **0.73** | **0.71** | 0.60 | 0.49 | 0.89 | 0.90 | 0.79 | 0.60 | 0.64 | 0.78 | 0.86 | 0.75 | 0.55 | **0.80** | 0.89 | 1.00 | **0.66** | 0.64 | 0.77 | 0.64 | **0.75** | 0.76 |
| Normal Direction | 0.51 | 0.60 | 0.12 | 0.62 | 0.75 | 0.10 | 0.10 | **0.85** | 0.42 | 0.12 | 0.10 | 0.60 | 0.45 | 0.23 | 0.09 | 0.60 | 1.00 | 0.30 | 0.51 | 0.00 | 0.20 | 0.22 | 0.71 |
| Screw Parameters [1] | 0.55 | 0.55 | 0.58 | 0.52 | 0.90 | 0.20 | 0.54 | 0.69 | 0.19 | 0.39 | 0.41 | 0.85 | 0.69 | 0.19 | 0.73 | 0.69 | 0.91 | 0.63 | **0.82** | 0.89 | **0.85** | 0.60 | 0.78 |
| DAgger E2E [31] | 0.32 | 0.63 | 0.10 | 0.36 | 0.19 | 0.39 | 0.12 | 0.13 | 0.00 | 0.12 | 0.38 | 0.10 | 0.09 | 0.00 | 0.00 | 0.10 | 0.25 | 0.20 | 0.17 | 0.02 | 0.00 | 0.00 | 0.00 |
| FlowBot3D (AF Only) [6] | 0.81 | 0.53 | 0.74 | 0.81 | 0.82 | 0.96 | **0.99** | 0.79 | 0.44 | 0.90 | **1.00** | 0.89 | 0.70 | 0.69 | 0.63 | 1.00 | 0.94 | 0.67 | 0.89 | 0.75 | 0.66 | 0.69 | 0.14 |
| AP Only (Ours) | 0.77 | 0.58 | 0.59 | **0.90** | 0.81 | 0.83 | 0.80 | 0.78 | 0.46 | **0.79** | 0.72 | 1.00 | 0.68 | 1.00 | 0.54 | 0.82 | 1.00 | 0.41 | 0.44 | 0.79 | 0.87 | 0.41 | 0.45 |
| **FlowBot++ (Ours - Combined)** | **0.82** | 0.70 | 0.62 | 0.89 | **0.91** | **0.96** | 0.96 | 0.83 | **0.72** | 0.78 | 0.85 | **1.00** | **0.76** | **1.00** | 0.68 | **1.00** | **1.00** | 0.43 | 0.63 | **0.91** | 0.81 | 0.45 | **0.82** |

Table 4: Success Rate Metric Results ($\uparrow$): Fraction of success trials (normalized distance less than 0.1) of different objects' categories after a full rollout across different methods. The higher the better.

## D   Metrics

We specify the metrics we used for our simulated experiments. First, shown in Table 1, we use Normalized Distance, which is defined as the normalized distance traveled by a specific child link through its range of motion. The metric is computed based on the final configuration after a policy rollout ($\mathbf{j}_{\text{end}}$) and the initial configuration ($\mathbf{j}_{\text{init}}$):

$$\mathcal{E}_{\text{goal}} = \frac{||\mathbf{j}_{\text{end}} - \mathbf{j}_{\text{goal}}||}{||\mathbf{j}_{\text{goal}} - \mathbf{j}_{\text{init}}||}$$

We also conduct experiments using the Success Rate: we define a binary success metric, which is computed by thresholding the final resulting normalized distance at $\delta$:

$$\text{Success} = \mathbb{1}(\mathcal{E}_{\text{goal}} \leq \delta)$$

We set $\delta = 0.1$, meaning that we define a success as articulating a part for more than 90%.

We show the success rate performance of our method and baselines in Table 4. Similar to the results using Normalized Distance, FlowBot++ outperforms previous methods.

# E    Simulation and Training Details

In simulation, the suction is implemented using a strong force between the robot gripper and the target part. During training step $t$, we randomly select an object from the dataset, randomize the object's configuration, and compute a new training example which we use to compute the loss using Eq. 7. During training, each object is seen in 100 different randomized configurations.

## E.1    Datasets

To evaluate our method in simulation, we implement a suction gripper in the PyBullet environment, which serves as a simulation interface for interacting with the PartNet-Mobility dataset [10]. The PartNet-Mobility dataset contains 46 categories of articulated objects; following UMPNet [12], we consider a subset of PartNet-Mobility containing 21 classes, split into 11 training categories (499 training objects, 128 testing objects) and 10 entirely unseen object categories (238 unseen objects). Several objects in the original dataset contain invalid meshes, which we exclude from evaluation. We train our models (FlowProjNet and baselines) exclusively on the training instances of the training object categories, and evaluate by rolling out the corresponding policies for every object in the dataset. Each object starts in the "closed" state (one end of its range of motion), and the goal is to actuate the joint to its "open" state (the other end of its range of motion). For experiments in simulation, we include in the observation $O_t$ a binary part mask indicating which points belong to the child joint of interest.

## E.2    Network Architecture

FlowProjNet, the joint Articulation Flow and Articulation Projection prediction model in FlowBot++, is based on the PointNet++ [34, 30] architecture. The architecture largely remains similar to the original architecture except for the output head. Instead of using a segmentation output head, we use a regression head. The FlowProjNet architecture is implemented using Pytorch-Geometric, a graph-learning framework based on PyTorch. Since we are doing regression, we use standard L2 loss optimized by an Adam optimizer [35].

The articulation type classifier model in FlowBot++, which is used to predict prismatic vs. revolute objects, is also based on the PointNet++ architecture. The architecture now uses a classification head, which outputs a global binary label representing the articulation label. We use standard Binary Cross Entropy loss optimized by an Adam optimizer [35] and we achieve 97% accuracy on test objects.

## E.3    Ground Truth Labels Generation

### E.3.1    Ground Truth Articulation Flow

We implement efficient ground truth Articulation Flow generation. At each timestep, the system reads the current state of the object of interest in simulation as an URDF file and parses it to obtain a kinematic chain. Then the system uses the kinematic chain to analytically calculate each point's location after a small, given amount of displacement. In simulation, since we have access to part-specific masks, the calculated points' location will be masked out such that only the part of interest will be articulated. Then we take difference between the calculated new points and the current time step's points to obtain the ground truth Articulation Flow.

### E.3.2    Ground Truth Articulation Projection

We also implement efficient ground truth Articulation Projection generation. For each object, the system reads the current state of the object of interest in simulation as an URDF file and parses it

to obtain the origin $v$ and direction $\boldsymbol{\omega}$ of the axis of articulation. The system then uses Eq. 2 to calculate the Articulation Projection label. Since we have access to part-specific masks in PyBullet, the calculated points' location will be masked out such that only the points on part of interest will be articulated.

## E.4  Using Segmentation Masks

As mentioned above, we use part-specific segmentation masks to define tasks. Specifically, we follow the convention in [12, 6, 13], where a segmentation mask is provided to give us the part of interest. Thus, it is possible that an object (e.g. a cabinet) could have multiple doors and drawers at the same time. By the construction of the dataset [12, 6, 13], in each data point (object), a mask is used to define which part on the object needs to be articulated. For the cabinet example, if the mask is provided for a drawer, then the cabinet is classified as a prismatic object. If the mask is provided for a door, then it is classified as a revolute object. We use segmentation masks in the following steps of the FlowBot++ pipeline:

1. **Articulation Flow Ground Truth**: During the generation of articulation flow labels, we use segmentation masks to mask out irrelevant parts on objects so that those parts' articulation flow values will be zeroed out. In this way, FlowProjNet will learn to predict all-zero on irrelevant parts.

2. **Articulation Projection Ground Truth**: Similar to how we use the mask in Articulation Flow Ground Truth generation, we only produces the projection vectors for the relevant masked points.

3. **Articulation Flow and Articulation Projection Prediction**: During the learning step of FlowProjNet, we use segmentation masks as an additional per-point channel into the network, where 1 represents relevant points and 0 represents irrelevant points. In this way, the network output learns to be conditioned on this extra channel such that it does not output values on irrelevant parts.

4. **Articulation Parameters Estimation**: When estimating $\boldsymbol{\omega}$ and $v$, we first obtain a per-point estimate. To make the estimate more robust, we aggregate all points on the relevant part, which is masked using the segmentation mask, and average them out to get a robust estimate.

## E.5  Hyperparameters

We use a batch size of 64 and a learning rate of 1e-4. We use the standard set of hyperparameters from the original PointNet++ paper.

# F  Real-World Experiments

## F.1  Experiments Details

We experiment with 6 different objects in the real world. Specifically, we choose 5 revolute objects: Oven, Fridge, Toilet, Trashcan, and Microwave, where the predicted trajectories are generated using Eq. 5, and we choose 1 prismatic object: Drawer, where the predicted trajectories are generated via Eq. 6. For each object in this dataset, we conducted 5 trials of each method. For each trial, the object is placed in the scene at a random position and orientation. For each trial, we visualize the point clouds beforehand and hand-label the segmentation masks using bounding boxes. We then pass the segmentation masks as the auxiliary input channel to FlowProjNet and use them to aggregate the final output to improve robustness. We then qualitatively assess the prediction by visualizing the points belonging to the segmented part under the predicted trajectory's transformation. We show some examples of the predictions in Fig. 8.

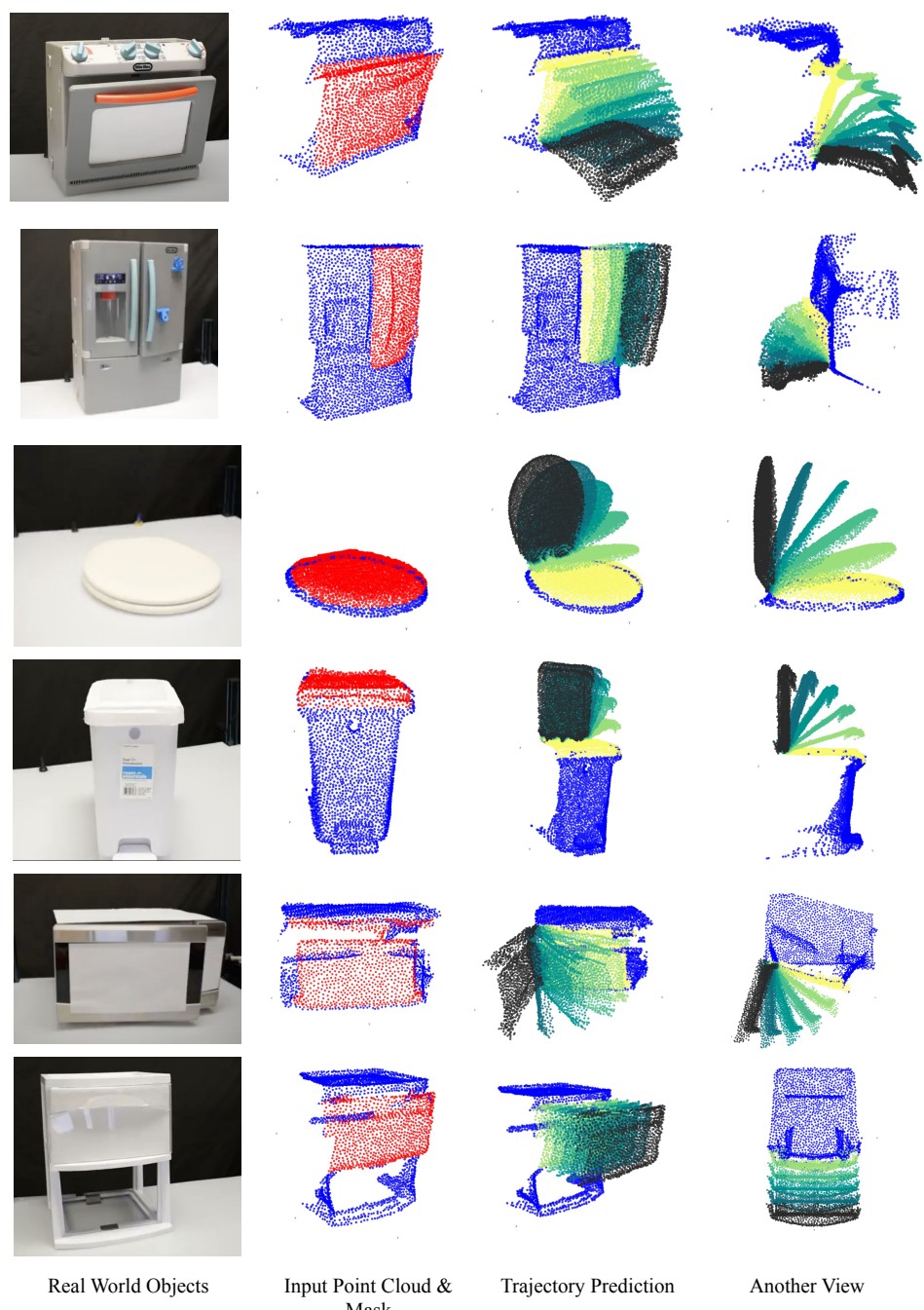

Real World Objects     Input Point Cloud & Mask     Trajectory Prediction     Another View

Figure 8: Successful Predictions on 6 Real-World Objects. We show FlowBot++'s prediction quality of 6 different real-world objects. For better readability, we provide an alternative view for each prediction. From top to bottom, the objects are: Oven, Fridge, Toilet, Trashcan, Microwave, and Drawer.

## F.2 Robotic System Implementation

We provide the details of the physical robot system implementation of FlowBot++. The setup remains straightforward and largely similar to previous works [28, 6].

### F.2.1 Hardware

In all of our real-world experiments, we deploy our system on a Rethink Sawyer Robot and the sensory data (point cloud) come from an Intel RealSence depth camera. The robot's end effector is an official Saywer Parallel-Jaw Gripper. We set up our workspace in a 1.2 m by 1.00 m space put together using the official Sawyer robot mount and a regular desk. We set up the RealSense camera such that it points toward the center of workspace and has minimal interference with the robot arm-reach trajectory.

### F.2.2 Solving for the Robot's Trajectory

The choice of the contact point is similar to the procedures described in Eisner et al. [6]. When the contact point's full trajectory is predicted using Eq. 5 or 6 based on the part's articulation type, the robot should just plan motions in order to make its end-effector follow the predicted trajectory. Once a successful contact is made, the robot end-effector is rigidly attached to the action object, and we then use the same predicted trajectory waypoints as the end positions of the robot end effector, and then feed the end-effector positions to MoveIt! to get a full trajectory in joint space using Inverse Kinematics. For *prismatic objects*, this is convenient because the robot gripper does not need to change its orientation throughout the predicted trajectory. For *revolute objects*, we propose a method to efficiently calculate the robot's orientations in tandem with the positions in the planned trajectory. Concretely, the trajectory of Eq. 5 gives us the end-effector's xyz positions in the trajectory; it is rigidly attached to the contact point, so we could treat their xyz positions to be the same in this trajectory. Now, we are interested in obtaining the orientation of the end-effector for each step. Assume the end-effector's orientation (obtained via Forward Kinematics, in the form of rotation matrix in $SO(3)$) is $\mathbf{q}_0$ when making a successful contact with the part of interest. By definition, each step in $\tau_{\text{revolute}}$ corresponds to a unique rotation matrix $\mathbf{R}(\phi_g/K)$, representing the difference of rotation due to the increase of opening angle in each step. We then calculate the robot end-effector's orientation at each step $i$:

$$\mathbf{q}_i = \mathbf{R}(\frac{\phi_g}{K})\mathbf{q}_{i-1} \tag{8}$$

by applying the difference of rotation onto the orientation's rotation matrix itself iteratively. Thus, in this case, the robot's end-effector's full $SE(3)$ trajectory is obtained:

$$\tau_{\text{ee}} = \left\{ \tau_{\text{revolute}}^i, \mathbf{q}_i \right\}_{\forall i \in [0,K]} \tag{9}$$

We then obtain the robot's joint-space trajectory using Inverse-Kinematics (IK):

$$\tau_{\text{joint}} = IK(\tau_{\text{ee}}) \tag{10}$$

## F.3 Reducing Unwanted Movements

With the ability to control the full 6D pose of the robot end-effector in the trajectory, we are also able to reduce the unwanted movements of the object itself. In FlowBot3D [6], the suction gripper's rotation is controlled using a heuristic based on the flow direction prediction, which is often off. Thus, incorrect rotation could cause the gripper to yank the object too hard in a wrong direction, causing unwanted motion of the articulated object or even detaching the gripper from the object surface. With the full 6D gripper trajectory produced by FlowBot++ as a byproduct, the relative pose between the gripper and the articulated part remains the same as when a contact is made throughout the trajectory. This largely eliminates the unwanted movement problem in [6]. We illustrate this property in Fig. 9. A disadvantage of deploying FlowBot3D in the real world is that each step is prone to error, causing the gripper to move to wrong directions, which would unexpectedly move the object, potentially causing damage. Using the full gripper trajectory derived from Eq. 9, FlowBot++

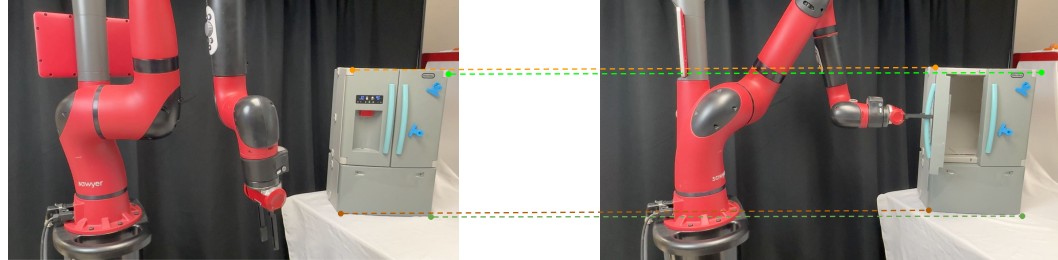

FlowBot++ (Ours)

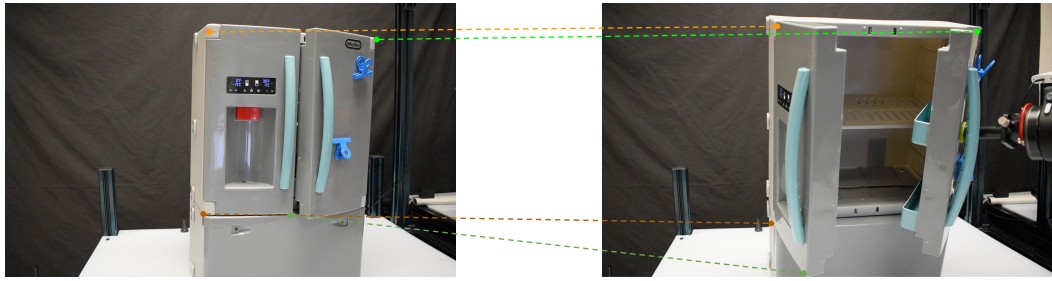

FlowBot3D

Figure 9: FlowBot++ Reducing Unwanted Movements of the Object. **Top:** FlowBot++ opens the left door of the fridge. The position and pose of the body of the fridge remain unchanged. **Bottom:** FlowBot3D opens the right door of the fridge. Due to wrong flow prediction at intermediate steps, the gripper yanks the fridge too hard that it tips over, causing unwanted motions of the fridge and opening the wrong door by accident.

is more likely to be more compliant with respect to the object's kinematic constraints without using hand-designed heuristics based on Articulation Flow predictions. The position and pose of the body of the articulated object are then able to remain unchanged. In contrast, FlowBot3D has more points of failure due to its closed-loop nature. When a single step's Articulation Flow prediction is off - namely, non-parallel to the ground-truth flow direction, the gripper would move against the object's kinematic constraint, moving the other parts of the object unexpectedly. Please note that this is better understood by watching the video comparisons on our website.

### F.4   Failure Case

We illustrate a failure case of FlowBot++ deployed in the real world here. The failure is caused by predictions that are off, which results in off-axis rotation. The failure case is shown in Fig. 10. In

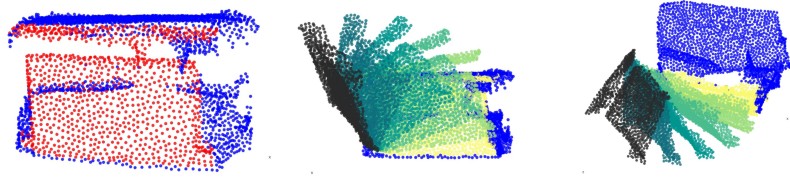

Figure 10: Failed FlowBot++ Prediction on a Microwave. Imperfect articulation parameter prediction caused the rotation to be off-axis.

this prediction, the prediction result in incorrect articulation parameters. From the visualization, the predicted axis is off, causing the rotated part to go "off the hinge." If a real robot were to execute this, the planned motion would either be infeasible or make the robot lose contact with the grasp point.

