# OpenReview forum: "FlowBot++: Learning Generalized Articulated Objects Manipulation via Articulation Projection"
_robot-learning.org/CoRL/2023/Conference — CoRL 2023 Poster_

### Official Review · Reviewer_frrv · 2023-07-14

**Confidence:** 4
**Originality:** Good
**Technical Quality:** Excellent
**Clarity Of Presentation:** Excellent
**Impact:** 3

**Recommendation:**

Weak Accept: I recommend accepting the paper, but will not argue for my recommendation if the majority of other reviewers have a different opinion.

**Review:**

Strengths
- This work is written very clearly.
- This work has strong motivation and execution. It identifies that the previous state-of-the-art method, FlowBot3D, lacks the ability to plan a manipulation trajectory, and proposes a a sensible representation, Articulation Projection, to overcome this limitation.
- This work has built a system that can be deployed in the real world.

Weaknesses
- Missing comparison with previous pose or motion prediction methods.
  One key contribution of this work is Articulation Projection, which seems to be a pure vision module that takes a segmented point cloud and outputs the motion axis. It seems that in principle, this module can be replaced with any other articulation motion prediction method, such as [1] and [17] discussed in related work. This work is missing a comparison with these baselines.
- It is not clear how incorrect segmentation and noisy point clouds in real-world scenario would affect the performance of this method.

**Quality Of The Limitations Section:**

Limitations are addressed clearly

**Questions For Rebuttal:**

Most importantly, I think this work needs to provide justifications for the proposed Articulation Projection by comparing with other motion prediction methods as I have discussed in the *review* section. Currently, the baseline comparison includes a Bullet-based simulator and manipulation policies, which cannot provide much insights about the performance of the vision module itself. Ideally, this work should show the effectiveness of Articulation Projection by including a baseline comparison where the input is a point cloud (preferably generated with a realistic depth camera simulator) with segmentation, and the output is the predicted motion axis.

**Robotics Focus:**

Sufficient demonstration on hardware

**Summary Of Paper:**

This work proposes Articulation Projection, a per-point representation of articulated motion. Building on top of this representation, this work proposes FlowBot++, a policy for manipulating articulated objects using 3D point cloud as input.

This work presents experiments showing that after trained in simulated environments, FlowBot++ can manipulate unseen articulated objects in both simulation and real-world.

**Summary Of Recommendation:**

This work follows a standard pipeline of first predicting articulated motion parameters and next plan trajectories to manipulate the articulated objects. This pipeline seems to have very limited novelty.
However, this work is also executed quite well and show clear improvement over the previous FlowBot3D method. It also implements a system to deploy the trained model in the real world. Therefore, I think it overall provides a valuable system for generalizable articulated object manipulation.

---

### Official Review · Reviewer_E3aG · 2023-07-19

**Confidence:** 5
**Originality:** Good
**Technical Quality:** Good
**Clarity Of Presentation:** Very Good
**Impact:** 4

**Recommendation:**

Weak Accept: I recommend accepting the paper, but will not argue for my recommendation if the majority of other reviewers have a different opinion.

**Review:**

The authors tackle an important problem and present a novel approach for addressing it. The paper is well-written and communicates the main ideas comprehensively. The authors motivate the problem and state their contributions clearly. Experiments, both in simulation and on real-world objects, provide evidence in support of the work. However, the manuscript requires additional work before I feel confident recommending the work for acceptance. Detailed feedback is as follows:
- All robot manipulation experiments consider tasks of opening/closing the objects to their full range of motion. Consider adding experiments for partially opening and closing these objects.
- Cite previous work in Line 123.
- Consider adding a brief justification to support the statement "... estimating the Articulation Flow results in more generalizable prediction than estimating the articulation parameters directly" in lines 145-146 for completeness.
- Please include empirical results supporting the statement: "Empirically, the Articulation Projection prediction is sometimes less accurate than the Articulation Flow prediction" (lines 180-181), as it influences a critical design choice for the proposed approach.
- Add details of the FlowProjNet architecture in the manuscript or in the appendix.
- Authors of [1] presented an updated version of the work to use screw parameters for articulation model estimation in [2], which reported significantly better results. Consider comparing with [2] for a more comprehensive comparison.
- Are the articulation parameters estimated in the previous iteration used as a prior for prediction in the successive calls during object manipulation? If not, adding them as a prior should help to improve the performance.
- Another interesting metric to demonstrate the effectiveness of the proposed approach would be to show the robot FT sensor data during manipulation experiments. If the predictions are accurate, experienced forces and torque should be low.
- Update Table 1 on the website.
- Typographical:
  - "how any" Line 124
  - "Both of both worlds" Line 182-183
  - Correct the spacing between lines 201-202

References:
[1] Jain, Ajinkya, et al. "Screwnet: Category-independent articulation model estimation from depth images using screw theory." 2021 IEEE International Conference on Robotics and Automation (ICRA). IEEE, 2021.
[2] Jain, Ajinkya, et al. "Distributional depth-based estimation of object articulation models." Conference on Robot Learning. PMLR, 2022.

**Quality Of The Limitations Section:**

Additional details required

**Questions For Rebuttal:**

1. All robot manipulation experiments consider tasks of opening/closing the objects to their full range of motion. Consider adding experiments for partially opening and closing these objects.
1. Please include empirical results supporting the statement: "Empirically, the Articulation Projection prediction is sometimes less accurate than the Articulation Flow prediction" (lines 180-181), as it influences a critical design choice for the proposed approach.
1. Can the proposed approach handle multiple joints? If not, how can the proposed method be extended to handle multiple joints?

**Robotics Focus:**

Sufficient demonstration on hardware

**Summary Of Paper:**

Articulated objects are ubiquitous in human environments. Interacting safely with them will be a necessity for service robots while assisting humans. The proposed method addresses this requirement and presents a novel approach to estimating articulation properties for objects without requiring prior interactions with them. The proposed method, FlowBot++, predicts dense per-point motion and articulation parameters for a provided segmented 3D point cloud of the object. Experiments demonstrate that FlowBot++ outperforms the SoTA methods in estimating motion and articulation parameters for novel objects and qualitatively establish that the estimated parameters are reasonably accurate to allow safe interactions with these objects.

**Summary Of Recommendation:**

The authors study an important problem and present a novel approach for solving it. The paper is well-written and effectively communicates the key concepts. The authors motivate the problem and concisely summarize their contributions. Experiments conducted both in simulation and on real-world objects demonstrate the effectiveness of the proposed approach. However, the manuscript would benefit from further refinement.

**Post-rebuttal update:**
I appreciate the authors' responses to my questions. However, I agree with other reviewers that the task completion rate alone does not provide strong evidence in favor of the efficacy of the proposed method, and adding experiments demonstrating the effectiveness of individual components would make the paper stronger. Hence, I would like to maintain my score.

---

### Official Review · Reviewer_8kbD · 2023-07-19

**Confidence:** 3
**Originality:** Very Good
**Technical Quality:** Good
**Clarity Of Presentation:** Very Good
**Impact:** 2

**Recommendation:**

Weak Accept: I recommend accepting the paper, but will not argue for my recommendation if the majority of other reviewers have a different opinion.

**Review:**

The paper is very clear and follows a logical flow making it a pleasure to read. The literature review is comprehensive. The reviewer especially likes the style of splitting into different category and at end of each highlighting the difference between this and aforementioned works.

Pros
The scheme is its able to generalize to objects with different articulation structures. By leveraging articulation projection, the model can transfer the learned knowledge to novel objects, even those with distinct joint configurations. This flexibility showcases the adaptability and versatility of the proposed framework.

Furthermore, the paper provides comprehensive evaluations and comparisons with existing state-of-the-art methods, demonstrating the superiority of FlowBot++ in terms of manipulation accuracy, robustness, and generalization capability. The results clearly indicate that the proposed approach outperforms other methods in challenging scenarios, where object articulation and complexity play significant roles.

Cons please see next section.


**Quality Of The Limitations Section:**

Additional details required

**Questions For Rebuttal:**

One major question for me is that the task that the robot is dealing with such as opening the oven does not really require explicit learning. We can do most of the task from a control perspective with an impedance controller. I would like the authors to comment on the advantages of the proposed scheme compare with control-based approaches.

**Robotics Focus:**

Sufficient demonstration on hardware

**Summary Of Paper:**

This paper presents a novel approach to generalized articulated object manipulation using the concept of articulation projection. The authors propose a novel framework that leverages deep learning techniques to enable robots to manipulate a wide range of objects with articulated parts.

**Summary Of Recommendation:**

It is a good work but I think most of the task here is already solved therefore I recommend a weak accept

---

### Official Review · Reviewer_uUTN · 2023-07-20

**Confidence:** 4
**Originality:** Very Good
**Technical Quality:** Excellent
**Clarity Of Presentation:** Excellent
**Impact:** 4

**Recommendation:**

Weak Accept: I recommend accepting the paper, but will not argue for my recommendation if the majority of other reviewers have a different opinion.

**Review:**

The authors observe that simply using articulation flow as a representation lacks the ability to predict a full, multi-step trajectory from the initial observation, hence the need for a per-point 3D representation that integrates motion flow and articulation parameters.

Strengths:
* Significant reduction in execution time compared to FlowBot3D (17.1s vs. 1.2s)
* Real world experiments with Sawyer robot: reliable predicted trajectory, and execution is much more smooth than FlowBot3D

Weaknesses:
* Could the authors provide an interpretation for why the proposed method fails in some categories when comparing with FlowBot3D (I.e. phone, laptop, button, etc.
* Could the authors provide an analysis of the failure cases, and ways to mitigate? Examples of failure cases were mentioned in the supplementary, however, it would be good to include an analysis as well
* Could the authors provide a comparison of the difference between objects in training vs. testing data? It appears that the objects used in real-robot experiments are toy versions, do they resemble closely what was seen during training?
* Generalizability: how well does the method generalize to unseen objects in the real world? How well does the method cope with input noise?

**Quality Of The Limitations Section:**

Limitations are addressed clearly

**Questions For Rebuttal:**

Overall, the paper was a pleasure to read. However, there are a few things which I'm hoping the authors could clarify, as summarized below (see section above for details):
* Additional interpretation for method's failure in certain categories compared to FlowBot3D (e.g., phone, laptop, button).
* Analyze and propose ways to mitigate failure cases, with examples from the supplementary materials.
* Compare objects in training and testing data, specifically real-robot experiment objects resembling trained objects.
* Evaluate method's generalizability to unseen real-world objects and its robustness to input noise.
 Further, there are some typos in the paper which should be fixed (i.e. L. 182 both --> best).

**Robotics Focus:**

Sufficient demonstration on hardware

**Summary Of Paper:**

This paper introduces FlowBot++, a 3D vision-based robotic system designed to understand and manipulate articulated objects like doors and drawers. Unlike previous approaches, FlowBot++ combines per-point motion and articulation parameters to achieve more accurate predictions. The system demonstrates successful articulation of various objects in simulated experiments using the PartNet-Mobility dataset and shows generalizability in real-world scenarios with real objects and a Sawyer robot. Overall, there are two main contributions: 1) per-point 3D representation of articulated objects (integrates motion under actuation and articulation parameters).
2) demonstrations on a robot manipulation system to manipulate novel articulated objects in a zero-shot manner.

**Summary Of Recommendation:**

The paper was enjoyable to read, and the authors have presented a promising approach. However, there are a few points that require clarification as mentioned above, with emphasis on providing detailed analysis on generalizability and training data. Addressing the listed points will strengthen the clarity of the paper.

---

### Decision · Program_Chairs · 2023-08-30

**Decision:**

Accept (Poster)

**Comment:**

The reviewers agree that the paper is well-written and enjoyable to read. I also do not find any critical issue in the paper. The reviewer raised a few points but I believe that the authors have answered them very clearly.

Some reviewers mentioned that either the impact will be small or the novelty is limited. This weakens the lowers quality of the paper. Even though all reviewers agree that the paper can be accepted, the average recommendation level is very high in this area and the paper is only borderline acceptable.

The paper is definitely robotics-oriented with good experiments on the real robot. All reviewers agree on this point.